# Association between physical activity duration and depressive symptoms in adolescents: A longitudinal study in a rural city in Japan

Toshinobu Kawai[1,2]*, Zentaro Yamagata[3,4]

**1** Institute of Health and Sport Sciences, University of Tsukuba, Tsukuba, Ibaraki, Japan, **2** Department of Integrated Applied Life Science, Integrated Graduate School of Medicine, Engineering and Agricultural Sciences, University of Yamanashi, Chuo, Yamanashi, Japan, **3** Department of Health Sciences, Faculty of Medicine, University of Yamanashi, Chuo, Yamanashi, Japan, **4** Center for Birth Cohort Studies, University of Yamanashi, Chuo, Yamanashi, Japan

* kawai.toshinobu.gp@u.tsukuba.ac.jp

**Data Availability Statement:** This dataset is sensitive data held by local governments that require consideration, such as the mental health status of children. The Ethics Committee of Koshu City and the University of Yamanashi School of

## Abstract

In Japan, physical activity duration in junior high schools is substantially higher than that in elementary schools. Using longitudinal data, this study examined the association between changes in physical activity duration and depressive symptoms in 1225 adolescents without depressive symptoms at baseline (51.2% female) from a rural city in Japan. Adolescents in the group that changed from "Active" (physical activity duration ≥7 h /week) in the fifth-grade of elementary school (age 11 years) to "Inactive" in the second-grade of junior high school (age 14 years) had a higher odds ratio of being rated as having depressive tendencies at age 14 years than adolescents in the Active–Active group. Additionally, the results of the cross-lagged effects model analysis suggested that depressive symptoms may be a factor in the reduced physical activity duration.

## Introduction

Recently, interest in depressive symptoms among adolescents has been increasing, including the conduct of many related surveys in Japan. In a survey of 3331 Japanese children and adolescents using the Birleson Depression Self-Rating Scale for Children (DSRS-C) [1] conducted in 2004, depressive symptoms exceeding the cutoff value were observed in 7.8% and 22.8% of elementary and junior high school students, respectively, which is higher among students in late elementary to junior high school [2]. Another 2016 survey of junior high school students reported that 20.5% of the students scored above the cutoff value set at 16 points [3]. These percentages were nearly constant irrespective of the time of measurement and are higher than those reported by surveys in Europe and the United States using a similar scale. For instance, Ivarsson et al. [4] found that 7.0% of Swedish adolescents (age 13–17) scored above the cutoff value of 15 points on the DSRS-C. This discrepancy is attributed to socioeconomic influences are the cause of this discrepancy [2].

Medicine have approved the research under their restrictions. Therefore, in principle, only information that has undergone statistical analysis has been made public. The email addresses of the responsible division are as follows: Clinical Research Support Group, General Administration Division, University of Yamanashi School of Medicine (Committee Secretariat) e-mail: rec-med@yamanashi.ac.jp The website of the University of Yamanashi Ethics Committee is also provided for reference. https://rinri.yamanashi.ac.jp/.

**Funding:** The author(s) received no specific funding for this work.

**Competing interests:** The authors have declared that no competing interests exist.

Depressive symptoms in adolescence include not only depressed mood, irritability, and decreased fun, but also sleep problems, appetite and weight changes, self-doubt, and suicidal thoughts [2]. These symptoms are associated with poor academic performance, decreased social well-being, and problematic social behaviors [5]. Furthermore, the presence of depressive symptoms in adolescents is reportedly predictive of both the intensity of depressive symptoms and the initial occurrence and subsequent recurrence of depression into late adolescence and early adulthood [6]. Thus, depressive symptoms in adolescence not only affect school and daily life but also decrease future quality of life.

Physical activity effectively reduces depressive symptoms and has been implemented as an alternative or complementary treatment for adolescents with depression, with some validation of its effectiveness [7, 8]. Even in children without pathological depressive symptoms, physical activity is closely associated with mental health status [9], and an active lifestyle is recommended as a prevention strategy for mental health challenges [10, 11]. Specifically, many countries recommend that young children and adolescents engage in physical activity for at least 60 min daily [12–14]. Guidelines set by the Japan Sports Association (JASA) recommended that children engage in a diverse range of physical activities. The recommended minimum "volume" of physical activity for children includes engaging in at least 60 min of activity per day, encompassing play that promotes physical activity, daily living activities, physical education, and participation in structured sports [10]. The World Health Organization (WHO) has also released recommendations suggesting that 5–17 year olds should engage in a total of at least 60 min of moderate to vigorous physical activity daily, three times a week, which should be include activities to strengthen muscle and bone [13].

Although much evidence exists on the benefits of physical activity for maintaining a better mental health status and treating depression, most of this evidence comes from cross-sectional studies. Additionally, only a few of those studies were based on longitudinal data [8, 9, 15]. Vella et al. [9] analyzed longitudinal data of Australian children aged 12 and 14 years to determine the association between sports participation and mental health (SDQ: Strength and Difficulties Questionnaire). The results revealed a bidirectional association between time spent participating in organized sports and overall mental health. This association may differ depending on social and cultural differences [8]; therefore, verifying whether the result is directly applicable to Japanese adolescents is necessary. Yasuda et al. [15] conducted a one-year follow-up study of Japanese children between the fourth-grade of elementary school and the second-grade of junior high school (10–14 years old) and examined the association between DSRS-C depression scores and physical activity duration within the same school year. They concluded that physical activity duration influenced the onset of new problematic depressive symptoms only in male students.

In Japan, major lifestyle changes occur during the transition from elementary school to junior high school. Many junior high school students in Japan participate in extracurricular sports activities known as "*bukatsu*." *Bukatsu* takes place after school on weekdays and weekends, sometimes before morning classroom hours. Students spend an average of 2–3 h daily engaging in physical activity for 5–6 days weekly [16]. While participation in *bukatsu* is not mandatory, a substantial number of students opt to engage in these activities. According to a survey by the Ministry of Education, Culture, Sports, Science, and Technology of Japan, it was estimated that 60–75% of all students participate in sports-related *bukatsu* such as tennis, basketball, football, baseball, and track and fileld, among others [16, 17]. Thus, significant changes in lifestyle and physical activity during the transition from elementary to junior high school may affect children's mental health; however, no studies have examined these associations using longitudinal data on physical activity-related lifestyle changes.

## Current study

Few studies have examined the association between physical activity and depressive symptoms using longitudinal data of adolescents in Japan. In particular, no studies have examined the association between changes in physical activity duration and changes in depressive tendencies during the period of significant lifestyle changes associated with the transition from elementary school to junior high school. This study aimed to examine the impact of changes in physical activity habits and other lifestyle factors during adolescence on depressive symptoms using longitudinal data of students in fifth-grade elementary school (age 10–11) to second-grade junior high school (age 13–14).

Previous studies have reported a negative association between physical activity and depressive symptoms and that physical activity interventions can improve depressive symptoms in children and adolescents with depression [7, 8]. Based on these findings, we hypothesized that an increase in physical activity between the fifth-grade of elementary school and the second-grade of junior high school would result in lower depressive symptoms, and a decrease in physical activity between the fifth-grade of elementary school and second-grade of junior high school would result in higher depressive symptoms.

## Methods

### Participants and procedure

We obtained and analyzed data from the cohort study called "Project Koshu," which was started in July 1988, as an administrative project for Koshu City, Yamanashi Prefecture (population approximately 30,000), and is still in operation. The project collects data at the time of pregnancy registration, at birth, at the time of infant health screening, and annually from the fourth-grade of elementary school to the third-grade of junior high school. Of these, this study included all 1890 children who were born between April 2, 2000 and April 1, 2006 and who responded at least once in the fifth-grade of elementary school (ages 10–11, hereafter referred as "11 y") and second-grade of junior high school (ages 13–14, hereafter referred to as "14 y") to the "Survey on Mental Health and Lifestyle Habits of Adolescence" conducted annually between June and July from 2011 to 2019 at all thirteen elementary and five junior high schools in Koshu City.

We utilized a retrospective analysis of the existing "Project Koshu" data to examine the adolescents' physical activity patterns and depressive symptoms from 11 y to 14 y. Concurrently, we followed the cohort prospectively from 11 y to 14 y to observe the progression of depressive symptoms and physical activity patterns as the adolescents transitioned through key developmental stages.

This study used data from the project conducted by the Koshu City Health Promotion Division and the Koshu City Board of Education. Ethical permission was obtained from the principal of the school where the study was conducted in the project. The school nurse or homeroom teacher at each school then gave a verbal explanation about the self-administered questionnaire to the children. Informed assent was obtained from all participants, and written informed consent was acquired from their legal guardians. Data were accessed by the corresponding author on August 8, 2022, with written permission by the mayor of Koshu City.

This study adhered to the ethical considerations as outlined in the Declaration of Helsinki. Ethical approval was obtained from the Ethics Committee of the Faculty of Medicine, University of Yamanashi (Approval No. 1694).

### Measures

**Depressive symptoms.** We assessed depressive symptoms using the Japanese version of the DSRS-C [1, 18] at two time points (11 y and 14 y). This scale has been evaluated for validity and reliability and has been used in many studies to assess depressive tendencies in adolescents

[1, 4, 18]. The scale has 18 questions, each of which is answered using a 3-point scale (0–2) with a total possible score of 0–36. Upon assessing Japanese children using the DSRS-C, a cut-off score of 16 points was found to yield higher specificity for identifying depressive symptoms than the original 15 points without compromising sensitivity [18]. Consequently, it has become standard practice to adopt a 16-point cutoff in such studies in Japan. This criterion was adhered to in the current study, with scores of <16 indicating "no depressive tendency" and ≥ 16 points indicating "depressive tendency." We excluded children aged 11 years with depressive tendencies from the analysis to exclude baseline effects (Fig 1).

**Physical activity duration.**   We assessed the physical activity duration of students at 11 y and 14 y using the question, "How many hours a week do you usually spend on physical activities outside of physical education (such as sports club activities, swimming school, and tennis school)?" based on the WHO Health Behavior in School-aged Children survey [19]. We simplified the question into two options: "1) about () hours per week in total" and "2) not at all." We set the "not at all" option to 0 h, and the number provided in the parentheses for the other option, the total weekly physical activity.

Thereafter, we divided the students into two groups based on physical activity duration and the JASA Physical Activity Guidelines for Children [10]: "Active" and "Inactive" groups comprising students who reported >7 and <7 h of physical activity weekly, respectively. Subsequently, the participants were classified into four groups, "Active–Active," "Active–Inactive," "Inactive–Active," and "Inactive–Inactive," based on the combination of their physical activity duration at the 11 y and 14 y.

**Fun of physical activity.**   We obtained responses for the following options: " fun," " somewhat fun," "undecided," "not much fun," and "not fun at all" Of these, we classified "fun" and "somewhat fun" as "fun," and the others as "not fun."

**Other moderator variables.**   We created a directed acyclic graph (DAG: S1 Fig) [20] based on the findings of previous studies and determined and included the moderator variables in the model. In the main analysis, we substituted body image [21, 22], internet addiction [23], sleep-onset status [24, 25], and social support [26, 27] for moderator variables.

Body image was assessed with the question, "Do you want to be thin or fat in terms of your body shape?" at 14 y. We classified the respondents into two groups: "want to be very thin" and "want to be a little thin" as "want to be thin;" and "want to stay the same," "want to be a little fat," and "want to be fat" as "do not want to be thin."

Internet addiction was assessed with the Young's Internet Addiction Test score [28] at 14 y. In this test, we classified scores of ≥40, assessed as "medium" or "high" for internet addiction, as "internet addictive," and scores of ≤ 39 or lower as "Not internet addictive."

Sleep-onset status was assessed with the question, "Can you fall asleep as soon as you get into bed?" at 14 y. We classified sleep-onset status into two groups: "falling asleep immediately" and " falling asleep in a short time" as "sleep well" and " not sleeping easily," "not sleeping until dawn," and "not sleeping" were classified as "not sleeping well."

Social support was assessed with the question, "Do you have anyone you can talk to about your problems and concerns?" at 14 y. We classified the respondents into two groups: the "Yes" group if they selected one or more of the following: "a family member," "a school teacher," "a friend," or "other," and the "No" group if they selected "no."

Based on the results of the DAG analysis, we analyzed breakfast skipping [29, 30] only in the univariate model and not as a moderator variable in the multivariate model.

## Statistical analyses

We performed descriptive analyses to determine the participant characteristics. We present the descriptive analysis results as means, standard deviations (SD), medians for continuous variables, and frequencies and percentages for categorical variables.

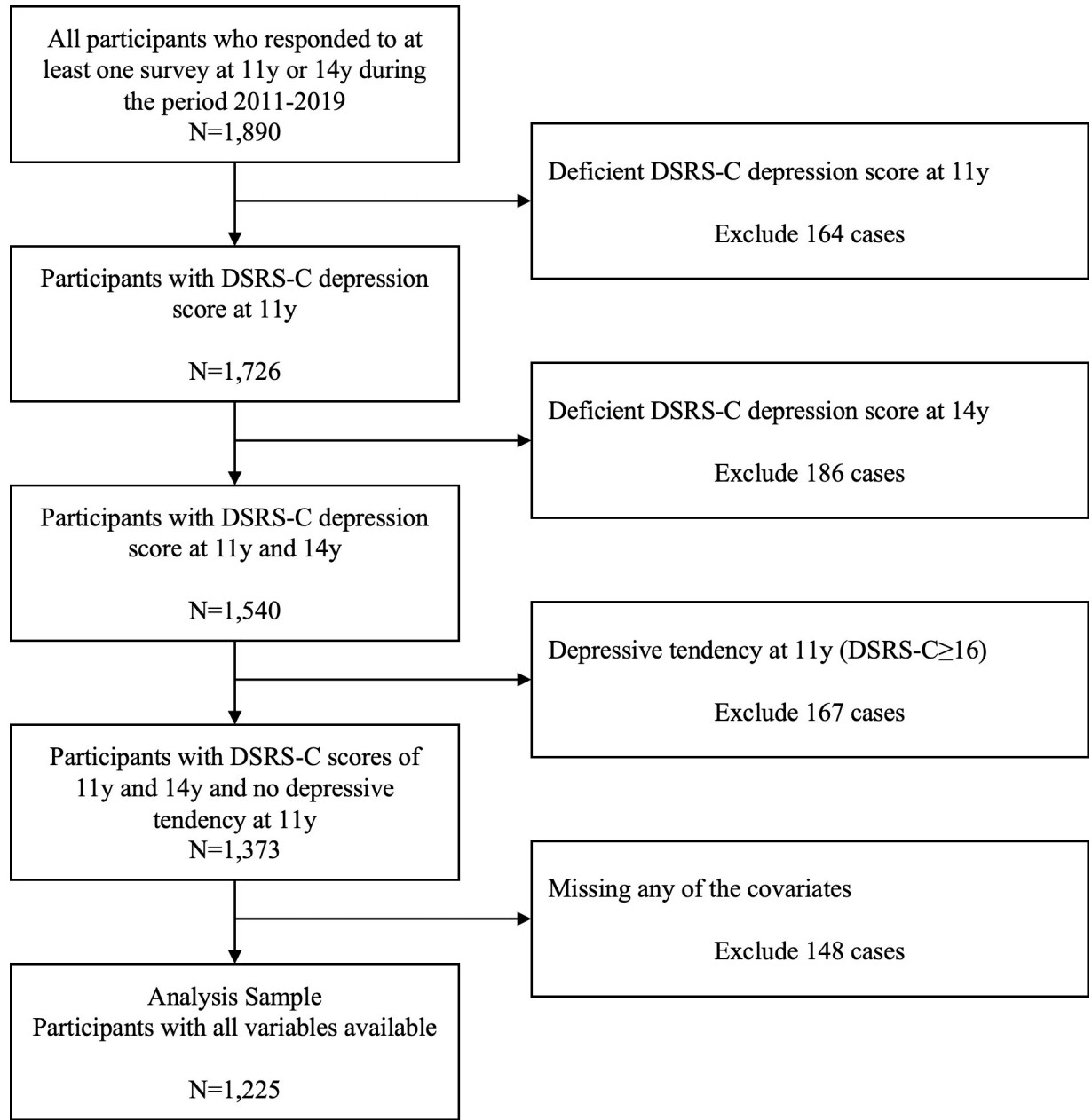

**Fig 1. Flow diagram of the participant selection process.** 11 y and 14 y, the fifth-grade of elementary school and the second-grade of junior high school, respectively. DSRS-C, the Birleson Depression Self-Rating Scale for Children.

We performed logistic regression analysis using "depressive tendency" at 14 y as the outcome variable for the main analysis. First, we analyzed univariate models for changes in physical activity duration and other moderator variables. We analyzed changes in physical activity duration using four groups of change patterns as dummy variables. Then, we analyzed the multivariate model incorporating "depressive tendency" at 14 y as an outcome variable, the changes in physical activity duration as an exposure variable, and other potentially confounding variables. In addition, we analyzed a model stratified by sex. Since there were few differences among participants' schools, multilevel analyses were not performed.

To examine a association between changes in depressive symptoms and changes in physical activity duration, we performed a sub-analysis using a cross-lagged effects model [31] and a model with all participants stratified by sex.

We used SPSS Statistics (version 29.0) and SPSS Amos (version 29.0) (IBM Corp., NY, USA) for the analysis.

## Results

This study included 1225 participants (598 males and 627 females) at two time points: the fifth-grade of elementary school (11 y) and the second-grade of junior high school (14 y) who responded to the study. All depressive symptom scores (DSRS-C), physical activity duration at the two time points, and all other moderator variables were included in the analysis. The participant selection process is illustrated in Fig 1.

The final analysis sample included 1225 participants, 51.2% of whom were females. The DSRS-C scores of all the participants were 6.79±3.96 and 7.53±5.32 at 11 y and 14 y, respectively. DSRS-C scores were higher for females than for males in both cases. We classified 8.9% of the participants as having depressive tendencies (DSRS-C $\geq$ 16) at 14 y, of which the percentage of females was higher than that of males. Table 1 shows the results of the detailed descriptive analysis of the participants.

Table 2 shows the change pattern from 11 y to 14 y in the physical activity duration and number of participants in each group. Inactive–Active was the most common pattern among all participants (37.8%), followed by Active–Active (32.4%), Inactive–Inactive (22.4%), and Active–Inactive (7.3%). Males tended to be more active than females.

Table 3 shows the results of the univariate logistic regression analysis of changes in physical activity duration and other moderator variables at 14 y with depressive tendency as the outcome. The association between changes in physical activity duration and depressive tendency was significantly higher among those in Active–Inactive (OR = 2.976 [95% CI: 1.498–5.913]) and Inactive–Inactive (OR = 1.890 [95% CI: 1.090–3.280]) groups when we used Active–Active group as reference. By sex, only Active–Inactive status (OR = 3.818 [95% CI: 1.398–10.431]) significantly increased the odds ratio for males. For other variables, the odds ratios for being female (OR = 1.725 [95% CI: 1.148–2.591]), not having fun with physical activity at 14 y (OR = 3.352 [95% CI: 2.159–5.204]), having a body image of "want to be thin" (OR = 2.036 [95% CI: 1.358–3.053]), being addicted to the Internet (OR = 2.652 [95% CI: 1.724–4.079]), not sleeping well (OR = 3.410 [95% CI: 2.003–5.803]), not having person to talk to (OR = 11.381 [95% CI: 4.980–26.014]), and being a breakfast skipper (OR = 3.284 [95% CI: 2.132–5.058]) were significantly high among the participants. By sex, the results were similar, except for body image and lack of fun with physical activity at 11 y.

Table 4 shows the results of the multivariate logistic regression analysis with depressive tendency at 14 y as the outcome variable, changes in physical activity duration as the exposure variable, and other variables as the moderator variables. Regarding the association between depressive tendencies and the pattern of changes in physical activity duration, the Active–Inactive pattern significantly increased the odds ratio when we used Active–Active as a reference for participants (OR = 2.441 [95% CI: 1.159–5.140]). In the univariate model, the Inactive–Inactive pattern of changes in physical activity duration, female sex, and body image were significant, while the odds ratio became lower or less significant in the multivariate model.

As the pattern of changes in physical activity duration affects depressive symptoms at 14 y, we analyzed the association between physical activity duration and depressive symptoms (DSRS-C scores) using a cross-lagged effects model [31] as a sub-analysis to examine the association between changes in physical activity duration and changes in depressive scores. Fig 2

**Table 1. Descriptive characteristics of the participants.**

| | Variables | Total | | Male | | Female | |
|---|---|---|---|---|---|---|---|
| | | **(n = 1225)** | | **(n = 598; 48.8%)** | | **(n = 627; 51.2%)** | |
| 11 y | | | | | | | |
| | Depression (DSRS-C) score | | | | | | |
| | Mean (SD) | 6.79 | (3.96) | 6.58 | (4.04) | 6.99 | (3.87) |
| | Median | 7.00 | | 6.00 | | 7.00 | |
| | Fun of physical activity, n (%) | | | | | | |
| | fun | 1158 | (94.5) | 580 | (97.0) | 578 | (92.2) |
| | not fun | 67 | (5.5) | 18 | (3.0) | 49 | (7.8) |
| | Physical activity duration, hrs:min | | | | | | |
| | Mean (SD) | 7:08 | (7:05) | 8:44 | (7:55) | 5:37 | (5:48) |
| | Median | 5:00 | | 6:00 | | 4:00 | |
| | Inactivity, n (%) | | | | | | |
| | active (PA≥7.0 h/week) | 487 | (39.8) | 298 | (49.8) | 189 | (30.1) |
| | inactive (PA<7.0 h/week) | 738 | (60.2) | 300 | (50.2) | 438 | (69.9) |
| 14 y | | | | | | | |
| | Depression (DSRS-C) score | | | | | | |
| | Mean (SD) | 7.53 | (5.32) | 6.64 | (5.02) | 8.38 | (5.46) |
| | Median | 6.00 | | 6.00 | | 7.00 | |
| | Depression tendency, n (%) | | | | | | |
| | no (DSRS-C <16) | 1116 | (91.1) | 558 | (93.3) | 558 | (89.0) |
| | yes (DSRS-C ≥16) | 109 | (8.9) | 40 | (6.7) | 69 | (11.0) |
| | Fun of physical activity, n (%) | | | | | | |
| | fun | 1052 | (85.9) | 548 | (91.6) | 504 | (80.4) |
| | not fun | 173 | (14.1) | 50 | (8.4) | 123 | (19.6) |
| | Physical activity duration, hrs:min | | | | | | |
| | Mean (SD) | 12:32 | (8:24) | 13:54 | (8:06) | 11:14 | (8:27) |
| | Median | 14:00 | | 15:00 | | 12:00 | |
| | Inactivity, n (%) | | | | | | |
| | active (PA≥7.0 h/week) | 860 | (70.2) | 463 | (77.4) | 397 | (63.3) |
| | inactive (PA<7.0 h/week) | 365 | (29.8) | 135 | (22.5) | 230 | (36.7) |
| | Body image, n (%) | | | | | | |
| | do not want to be thin | 656 | (53.6) | 423 | (70.7) | 233 | (37.2) |
| | want to be thin | 569 | (46.5) | 175 | (29.3) | 394 | (62.8) |
| | Internet addiction, n (%) | | | | | | |
| | no internet addictive | 1014 | (82.8) | 497 | (83.1) | 517 | (82.5) |
| | internet addictive | 211 | (17.3) | 101 | (16.9) | 110 | (17.6) |
| | Sleep-onset status, n (%) | | | | | | |
| | sleep well | 1131 | (92.4) | 556 | (92.9) | 575 | (91.7) |
| | not sleep well | 94 | (7.6) | 42 | (7.0) | 52 | (8.4) |
| | Having a person who talks to, n (%) | | | | | | |
| | yes | 1201 | (98.0) | 584 | (97.7) | 617 | (98.4) |
| | no | 24 | (2.0) | 14 | (2.3) | 10 | (1.6) |
| | Breakfast skipping, n (%) | | | | | | |
| | never | 1037 | (84.7) | 515 | (86.1) | 522 | (83.3) |
| | occasionally or every morning | 188 | (15.3) | 83 | (13.8) | 105 | (16.7) |

PA, Physical activity; DSRC-C, Birleson Depression Self-Rating Scale for Children.

**Table 2. Change pattern from 11 y–14 y in the physical activity duration and number of participants in each group.**

| Change pattern of the physical activity duration* | Total (n = 1225) | | Male (n = 598; 48.8%) | | Female (n = 627; 51.2%) | |
|---|---|---|---|---|---|---|
| Active–Active | 397 | (32.4%) | 251 | (42.0%) | 146 | (23.3%) |
| Active–Inactive | 90 | (7.3%) | 47 | (7.9%) | 43 | (6.9%) |
| Inactive–Active | 463 | (37.8%) | 212 | (35.5%) | 251 | (40.0%) |
| Inactive–Inactive | 275 | (22.4%) | 88 | (14.7%) | 187 | (29.8%) |

* Physical activity duration is defined as "Active" at least 7 h per week.

shows the results. Among all the participants, the path from depressive symptoms to physical activity duration was significant (β = -0.114, p<0.0001, GFI = 1.000, CFI = 1.000). This path was also significant by sex (Males: β = -0.113, p<0.001, GFI = 1.000, CFI = 1.000; Females: β = -0,116, p<0.001, GFI = 1.000, CFI = 1.000).

## Discussion

In this study, using longitudinal data from adolescents, we examined the effect of changes in physical activity habits on depressive symptoms due to transitioning from elementary to junior high school, in association with other lifestyle habits that affect depressive symptoms. The results showed that the group that changed to Active–Inactive had a higher odds ratio for having a depressive tendency at 14 y than the Active–Active group, whose physically active duration was >7 h weekly in both the fifth-grade of elementary school (11 y) and second-grade of junior high school (14 y). This association was significantly stronger in males and weaker in females (Tables 3 and 4). Moreover, in females, no significant association with depressive symptoms was found for other patterns of change in physical activity duration. These results were consistent in both the univariate and multivariate models. Analysis of the cross-lagged

**Table 3. Association between depressive symptoms in the second-grade of junior high school (14 y) and the changes in the physical activity duration or other covariates of the univariate model analysis.** The changes in physical activity duration were analyzed simultaneously with the group as a dummy variable.

| Change pattern of the physical activity duration* | Total (n = 1225) | | | Male (n = 598) | | | Female (n = 627) | | |
|---|---|---|---|---|---|---|---|---|---|
| | OR | (95% CI) | | OR | (95% CI) | | OR | (95% CI) | |
| Active–Active | reference | | | reference | | | reference | | |
| Active–Inactive | **2.976** | **(1.498** | **– 5.913)** | **3.818** | **(1.398** | **– 10.431)** | 2.155 | (0.837 | – 5.546) |
| Inactive–Active | 1.330 | (0.788 | – 2.246) | 1.543 | (0.685 | – 3.474) | 0.997 | (0.498 | – 1.994) |
| Inactive–Inactive | **1.890** | **(1.090** | **– 3.280)** | 2.182 | (0.848 | – 5.615) | 1.322 | (0.655 | – 2.670) |
| Covariates | | | | | | | | | |
| Sex: Female | **1.725** | **(1.148** | **– 2.591)** | - | - | - | - | - | - |
| 11 y fun of physical activity: not fun | 1.877 | (0.929 | – 3.790) | **4.317** | **(1.352** | **– 13.790)** | 1.141 | (0.467 | – 2.787) |
| 14 y fun of physical activity: not fun | **3.352** | **(2.159** | **– 5.204)** | **3.661** | **(1.633** | **– 8.208)** | **2.874** | **(1.685** | **– 4.901)** |
| Body image: want to be thin | **2.036** | **(1.358** | **– 3.053)** | 1.673 | (0.866 | – 3.233) | **1.932** | **(1.089** | **– 3.428)** |
| Internet addiction: addictive | **2.652** | **(1.724** | **– 4.079)** | **2.572** | **(1.277** | **– 5.177)** | **2.707** | **(1.561** | **– 4.693)** |
| Sleep-onset status: not sleep well | **3.410** | **(2.003** | **– 5.803)** | **2.559** | **(1.008** | **– 6.494)** | **3.911** | **(2.017** | **– 7.584)** |
| Not having a person to talk to | **11.381** | **(4.980** | **– 26.014)** | **12.132** | **(3.983** | **– 36.951)** | **13.190** | **(3.625** | **– 48.000)** |
| Breakfast skipping: occasionally or every morning | **3.284** | **(2.132** | **– 5.058)** | **2.940** | **(1.430** | **– 6.043)** | **3.395** | **(1.969** | **– 5.855)** |

OR, odds ratio; CI, confidence interval.

* Physical activity duration is defined as "Active" at least 7 h per week.

**Table 4. Association between depressive symptoms in the second-grade of junior high school (14 y) and the changes in physical activity duration and other covariates of the multivariate model analysis.**

| Change pattern of the physical activity duration* | Total (n = 1225) OR | (95% CI) | | Male (n = 598) OR | (95% CI) | | Female (n = 627) OR | (95% CI) | |
|---|---|---|---|---|---|---|---|---|---|
| Active–Active | reference | | | reference | | | reference | | |
| Active–Inactive | **2.441** | **(1.159** | **–** | **5.140)** | **3.158** | **(1.044** | **–** | **9.551)** | 1.822 | (0.653 | – | 5.085) |
| Inactive–Active | 1.161 | (0.668 | – | 2.020) | 1.580 | (0.675 | – | 3.702) | 0.909 | (0.437 | – | 1.890) |
| Inactive–Inactive | 1.148 | (0.618 | – | 2.133) | 1.362 | (0.482 | – | 3.851) | 0.954 | (0.434 | – | 2.098) |
| Gender: Female | 1.424 | (0.893 | – | 2.271) | – | – | | – | – | – | | – |
| 11 y fun of physical activity: not fun | 1.091 | (0.503 | – | 2.364) | **3.775** | **(1.087** | **–** | **13.112)** | 0.646 | (0.243 | – | 1.717) |
| 14 y fun of physical activity: not fun | **2.223** | **(1.316** | **–** | **3.755)** | 1.823 | (0.689 | – | 4.818) | **2.648** | **(1.408** | **–** | **4.981)** |
| Body image: want to be thin | 1.447 | (0.918 | – | 2.281) | 1.251 | (0.609 | – | 2.567) | 1.538 | (0.833 | – | 2.840) |
| Internet addiction: addictive | **1.987** | **(1.239** | **–** | **3.187)** | **2.202** | **(1.012** | **–** | **4.792)** | **1.919** | **(1.041** | **–** | **3.538)** |
| Sleep-onset status: not sleep well | **2.471** | **(1.385** | **-** | **4.408)** | 1.910 | (0.673 | - | 5.417) | **2.826** | **(1.378** | **-** | **5.797)** |
| Not having a person to talk to | **10.759** | **(4.283** | **-** | **27.025)** | **12.451** | **(3.604** | **-** | **43.015)** | **10.531** | **(2.554** | **-** | **43.430)** |

OR, odds ratio; CI, confidence interval.

* Physical activity duration is defined as "Active" at least 7 hours per week.

effects model for physical activity duration and DSRS-C showed that the path from depression score to physical activity was significant (Fig 2).

## Participant characteristics

In this study, to exclude the influence of depressive symptoms at baseline, we excluded 10.8% (= 167/1540) children with DSRS-C scores of >16 points at 11 y, those with depressive tendencies at the participant selection step (Fig 1). Although this percentage is slightly higher than that reported for other Japanese children of the same age (9.3%) [2], this group has the average characteristics of Japanese children. The DSRS-C score at 14 y, excluding children who have a depressive tendency at 11 y, was 7.53±5.32, which was lower than the previous reports of 10.61 ±6.67 [2] and 12.08±6.06 [31] in the same age group in Japan.

The percentage of participants with depressive tendencies at 14 y was 8.9%, which was significantly lower than the 23.3% reported by Denda et al. [2] and 25.1% reported by Kawakatsu et al. [32]. Although not shown in the results of this study, the depressive tendencies of the overall number of participants, before excluding participants with depressive tendencies at 11 y, was 11.7% (n = 1540), which is lower than that of a previous study on the same age group in

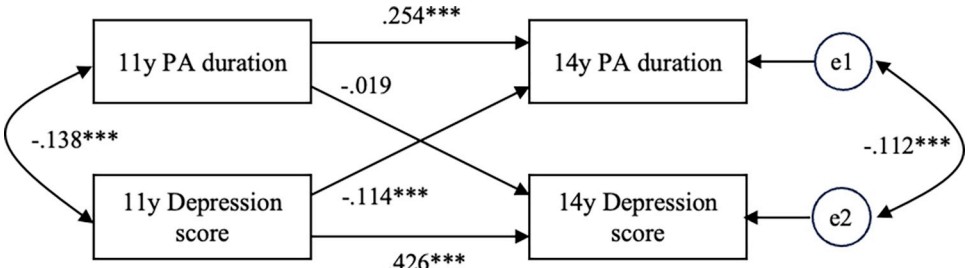

**Fig 2. Standardized regression coefficients (β) in cross-lagged effects model in the whole participants (n = 1225).** 11 y and 14 y, the fifth-grade of elementary school and the second-grade of junior high school, respectively. PA means physical activity. The depression score was derived from Birleson Depression Self-Rating Scale for Children (DSRS-C). "e" means error. Goodness of fit in this model is GFI = 1.000, CFI = 1.000. *** p<0.001.

Japan. These results indicated that the mental health of the participants in this study at 14 y was better than the average for Japanese children in the same age group. Given the numerous factors that influence depressive symptoms in adolescents and the complex nature of their associations, providing a structural explanation for the observed lower depressive tendencies among participants in the current study compared with the general Japanese adolescent population is challenging. However, we postulate that the social environments and physical activity levels in Koshu City may have played a role in reducing depressive tendencies.

Increased physical activity duration at 14 y compared to that at 11 y was different from the results of studies conducted on youth in England and Australia [9, 11]. This may be because many junior high school students in Japan participate in extracurricular sports club activities (*bukatsu*). According to the 2019 statistics of the Japanese government [16], the participation rate of junior high school students in *bukatsu* throughout Japan was 75.6% for males and 57.3% for females. The participation rates for junior high school students in Yamanashi Prefecture, where the participants in this study reside, were higher at 78.6% for males and 64.6% for females, indicating that junior high school students in this region have more opportunities to engage in physical activity compared to the average Japanese children. As for the national average physical activity duration, male junior high school students reportedly spent 817 min weekly (13.62 h/week) and females spent 596 min weekly (9.93 h /week) [16] on physical activity, whereas the average physical activity duration of the participants in this study was 13.90 h/week for males and 11.23 h/week for females. The duration of physical activity for males was almost the same as the national average, while that for females was longer than the national average. Although the exact rate of *bukatsu* participation among the participants in this study is unknown, assuming that it is similar to that of the region (Yamanashi Prefecture), the high rate of participation in *bukatsu* may be related to the higher physical activity duration of females relative to the national average.

## Association between physical activity and depressive symptoms

Interesting results were obtained regarding how changing patterns in physical activity duration affected depressive symptoms, which was the main concern of this study. To facilitate a comprehensive understanding of our findings, we first examined the results from the sub-analysis before exploring those from the main analysis.

The results of the cross-lagged effects model conducted during the sub-analysis showed a consistently significant association between depressive symptoms and physical activity duration, both in all participants and separately by sex. The results showed a reverse association with the assumptions of our model based on several previous studies. In other words, increased physical activity duration did not decrease depressive symptoms. However, the worsening of depressive symptoms decreased physical activity duration. Many previous cross-sectional studies have examined physical activity as an alternative treatment for adolescents with depression or the association between physical fitness level, physical activity, and mental health, concluding that increased physical activity improves depressive symptoms [7, 8]. While few studies have examined these associations based on longitudinal data from children with good mental health, our results are the first to show that, for children with good mental health, worsening mental health decreases physical activity duration.

Vella et al. [9], one of the few longitudinal studies we recognized, used a cross-lagged effects model to analyze the association between sports participation and mental health in Australian children of approximately the same age as our study. The study reported that overall psychological difficulties and internalizing problems (social and emotional problems), as assessed by parents using the Strength and Difficulties Questionnaire (SDQ), had a bidirectional

association with sports participation, whereas externalizing problems (conduct problems and inattention/hyperactivity problems) had only a negative association with subsequent sports participation. Although the difference in the assessment scale between the study and our study precludes direct comparison, the difference in participant depressive symptoms at baseline may have caused the differences in association with physical activity duration. Our study analyzed the cross-lagged effects model using data that excluded children with depressive tendencies at baseline. Therefore, the path by which physical activity duration reduces depressive symptoms was possibly underestimated in our study because of the floor effect. It is also possible that we overlooked the possibility of reducing depressive symptoms by increasing physical activity in children with depressive symptoms at baseline. However, Vella et al. [9] did not perform such baseline manipulations.

These differences may be attributed to Japan's sociocultural characteristics. In other words, participation in *bukatsu*. A certain number of children felt stressed by this activity [17, 33], and it is possible that the results of the sub-analysis were distorted by the children's participation in long hours of physical activity that was not of their own volition.

These may also explain the differences between the results of a few longitudinal studies conducted in Japan [15]. A one-year follow-up study of non-depressed (DSRS-C<16) children from the fourth-grade of elementary school (ages 9–10) to the second-grade of junior high school (ages 13–14) found that among males, a higher amount of physical activity reduced the risk of depressive symptoms. However, this study was a one-year follow-up at each grade level and did not consider lifestyle changes in physical activity between elementary and junior high schools. We believe that the differences between the results of the study and our study may support the idea that lifestyle changes affect depressive symptoms.

Since this study did not define the intensity of physical activity in the questions regarding physical activity duration, and thus judgments were based on individual subjectivity, it is unclear how participants perceived low-intensity physical activities, such as activities of daily living. Therefore, the actual amount of physical activity might have been underestimated. Although the Health Behavior in School-aged Children Study (HBSC) questionnaire, which was used as a reference in the development of the questionnaire for this study, was confirmed to be valid and reliable [19, 34, 35], it overestimated moderate-to-vigorous physical activity compared with an accelerometer and reported sexual differences in the accuracy of responses [35]. These factors may have influenced the result in the subanalysis.

Considering these factors, the results of this study may have limited application to Japanese children without depressive tendencies at 11 y. With these assumptions in mind, we discussed the results of our main analysis.

## Association between changes in physical activity habits and depressive symptoms

The main analysis of this study, which involved the examination of the association between change patterns in physical activity duration and depressive tendencies, is an examination of whether depressive tendencies change when physical activity increases or decreases over a relatively long period.

The results of the multivariate model using moderator variables showed that the odds ratio for depressive tendencies was high only when children who were active at 11 y became inactive at 14 y (Table 4). The association differs with sex; we observed a stronger and more significant association in males and a less significant association in females. In addition, the odds ratios for depressive tendencies increased significantly in all participants and differed with sex for those having no one to talk to at 14 y. Furthermore, the odds ratio increased when physical

activity was not at 14 y. We interpreted these results not as contradicting the results of the sub-analysis, but as a variety of stressors exacerbated depressive symptoms, physical activity duration decreased, and participants no longer felt fun when they were physically active. Japanese junior high school students are exposed to much stress in school life, including participating in *bukatsu* and attending "*juku*" (private after-school classes) for high school entrance examinations [17, 33]. At this time, the absence of someone to talk to can further worsen depressive symptoms.

Sports club activities have a significant impact not only on physical activity but also on social interactive activities, such as communication among members [36]. Good communication increases self-esteem and alleviates depressive symptoms [37]. The experience of social isolation due to COVID-19 has provided an opportunity to reconfirm the importance of human connections in adolescence for emotional stability [27]. This tendency is reportedly prominent among females. The effect of physical activity on females may be less than that on males, as shown by the results of this study, and interpersonal connections have a stronger influence on depressive symptoms among females. This result is supported by a one-year follow-up study of children and adolescents in the same area, which found no association between the amount of physical activity and depressive symptoms among females [15].

## Strength and limitations

This study was conducted in schools with high admission rates for each year. Overall, 96.4%–98.6% of pupils or students registered in each school; therefore, the cohort was considered to adequately reflect the situation of children in the target area. In addition, the baseline depressive symptoms and physical activity duration of the participants were similar to that of the average Japanese, which reflects the actual situation of children in Japan.

The findings of this study may not apply to children with depressive tendencies because they were excluded from the study at baseline (11 y). Moreover, regarding the changes in DSRS-C scores by age, a study conducted in Sweden in 1997 focusing on adolescents aged 13–18 years found no significant age-related differences in scores [38]. However, the potential for age-related changes to function as confounding factors cannot be entirely dismissed [32]. In the current study, data indicating depressive tendencies at 11 y were excluded; hence, the impact of this variable has not been examined. In addition, the findings of this study are likely influenced by the sociocultural background of Japan and may not apply to non-Japanese children.

This study was based on a self-administered questionnaire. There were individual differences in the understanding and interpretation of the questions. The validity and reliability of the results are considered limited.

## Conclusion

This study revealed that in male adolescents with good mental health, a pattern of change from active to inactive in terms of physical activity due to lifestyle changes increased the risk of moving from no depressive tendency to increased depressive tendency. This association was weakened in females, suggesting that the influence of social support, such as communication with friends and family, on depressive symptoms was stronger in females than in males. In adolescents with good mental health, the results suggest that changes in depressive symptoms may be a factor in the decrease in physical activity duration. Concurrently, the Japanese system of "*bukatsu*," an extracurricular sports club activity, may have influenced the results of this study. This study suggests that *bukatsu* positively affects depressive symptoms. Participation in *bukatsu* increases physical activity duration, and if a club has a positive atmosphere, communication with members may provide social support, decreasing depressive symptoms.

Although *bukatsu* prevented the general decrease in physical activity duration and increase in sedentary behavior duration in elementary school students transitioning to junior high school [11], it can also worsen depressive symptoms and decrease physical activity duration if mismanaged or due to poor relationship among the members. Additional investigations are needed to explore the association between club members' interpersonal dynamics, physical activity duration, and depressive symptoms in the context of *bukatsu* management.

This study provides new insights into the association between physical activity and depressive symptoms among young males, who have good mental health, the majority in Japnan adolescents. Further research is needed to determine whether this finding is applicable to young males or females of the same age group living outside of Japan with different sociocultural backgrounds.

## Supporting information

**S1 Fig. Changes of physical activity (PA) duration, depressive symptoms, and covariates Directed Acyclic Graph (DAG).** We made the DAG using the online resource "DAGitty" (https://www.dagitty.net/). We derived the variables and their relationships included in this DAG from previous studies and discussions with the co-author.
(TIF)

## Acknowledgments

The authors are deeply grateful to the Health Promotion Division and Board of Education of Koshu City and all elementary and junior high schools in Koshu City for their cooperation during data collection. We would like to thank Editage (www.editage.jp) for English language editing.

## Author Contributions

**Conceptualization:** Toshinobu Kawai, Zentaro Yamagata.

**Formal analysis:** Toshinobu Kawai.

**Methodology:** Toshinobu Kawai, Zentaro Yamagata.

**Supervision:** Zentaro Yamagata.

**Writing – original draft:** Toshinobu Kawai.

**Writing – review & editing:** Zentaro Yamagata.

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
