## [Decision Letter · Decision Letter 0]

9 Feb 2024

PONE-D-23-34588Association between physical activity time and depressive symptoms in adolescents: A longitudinal study in a rural city in JapanPLOS ONE

Dear Dr. Kawai,

Thank you for submitting your manuscript to PLOS ONE. After careful consideration, we feel that it has merit but does not fully meet PLOS ONE’s publication criteria as it currently stands. Therefore, we invite you to submit a revised version of the manuscript that addresses the points raised during the review process.

We look forward to receiving your revised manuscript.

Kind regards,

Mukhtiar Baig, Ph.D.

Academic Editor

PLOS ONE

2. We notice that your supplementary figures are uploaded with the file type 'Figure'. Please amend the file type to 'Supporting Information'. Please ensure that each Supporting Information file has a legend listed in the manuscript after the references list.

Reviewers' comments:

Reviewer's Responses to Questions

**Comments to the Author**

1. Is the manuscript technically sound, and do the data support the conclusions?

Reviewer #1: Yes

Reviewer #2: Yes

Reviewer #3: Yes

2. Has the statistical analysis been performed appropriately and rigorously? 

Reviewer #1: Yes

Reviewer #2: Yes

Reviewer #3: I Don't Know

3. Have the authors made all data underlying the findings in their manuscript fully available?

Reviewer #1: No

Reviewer #2: Yes

Reviewer #3: Yes

4. Is the manuscript presented in an intelligible fashion and written in standard English?

Reviewer #1: Yes

Reviewer #2: Yes

Reviewer #3: Yes

5. Review Comments to the Author

Reviewer #1: The manuscript has been carefully reviewed, and I have no specific comments to offer at this time. The authors have effectively presented their objectives, provided a relevant introduction, and conveyed their results with clarity. The overall structure of the manuscript is well-executed.

Should there be any additional information or clarification required, I will be happy to reconsider and provide feedback upon further review. Overall, the manuscript appears to be well-prepared and meets the standards for publication.

Reviewer #2: The article clearly articulates its purpose, providing a comprehensive understanding of the importance of physical activity."The objectives of the study are well-defined, making it easy for readers to grasp the focus of the research.

The article addresses ethical considerations adequately, ensuring participant confidentiality and research integrity.

This article significantly contributes to the existing literature on physical activity by [highlighting a unique aspect, presenting novel findings.

"The study fills a gap in the current knowledge, offering valuable insights that can inform future research and interventions."

The data analysis is thorough, and the results are presented in a clear and concise manner.

The interpretation of findings is insightful, linking results back to the broader implications for public health or specific populations.

The writing is engaging and accessible, making the content suitable for a wide range of readers.

The article balances academic rigor with readability, ensuring accessibility without sacrificing depth.

The article is well-supported by a robust selection of references, demonstrating a comprehensive review of existing literature."

The citations are relevant and appropriately used to support the arguments and findings presented.

Reviewer #3: Abstract:

Introduction:

1. The introduction provides a comprehensive background on depression in adolescents. In line 37, the included study with 3331 children was conducted in 2004, and the other research (line 41) was conducted in 2016. It would be better to connect this information as a trend. This will enable the reader to comprehend the trend of depression over the years.

2. The percentage of depression in the DSR-C studies was mentioned. In line 43, the percentages of depression were more significant in Europe and the USA. However, providing more specific information by including figures would be helpful. This would contextualise the prevalence of depression locally and worldwide.

3. The symptoms of depression were mentioned in the second paragraph—however, the sentence in line 50 needs paraphrasing to deliver the intended point.

4. In this third paragraph, the recommended duration of physical activity is documented. However, it would also be better to mention the recommended exercise type. (frequency and intensity). This will provide a whole picture of the physical activity recommendation.

5. In the last paragraph, bukatsu sport was mentioned. It would be helpful to briefly describe this sport and say if it is optional or mandatory for the students. This will keep the reader on track.

Methods:

1. This study includes massive and structured data, making the results more reliable. However, it was initially challenging to comprehend whether the study was retrospective or both retrospective and prospective. For example, In the 3rd paragraph, the author explains that a self-administered questionnaire was given to the participants. Was this done by the Koshu project or from your side by interviewing the subjects? Please clarify these points to be easier for the reader to follow.

2. In the participants and procedure, the data were collected from the Koshu project. Out of these, 1890 were selected. Does this represent all the available data, or was a sample size calculated? Please specify this information to comprehend the methodology.

3. Under the section depressive symptoms, some of the studies used 15 as a cut-off point. Please provide a more substantial justification if possible. This might underestimate the depressive symptoms, and it would be difficult to compare it with other international studies.

4. Does the physical activity time mean the duration or the timing? In the methods, the question refers to the duration; however, the table refers to the timing, as there is another question for the activity based on the duration spent in the physical activity. The world time is, therefore, overlapping between the timing and the duration.

5. Figure 1 shows the recruitment done in this study, which is a great visual explanation. However, it must be self-explanatory to make it easier for the reader to follow.

6. Table 1 divides body image into wanting to be fat, wanting to stay the same and wanting to be thin. However, the text line 172 mentioned that it would be classified into two categories: want to be thin and don’t want to be thin. Please change the results in the table to be consistent.

7. Table 1, including physical activity time. Please specify if it is referred to as timing or duration, as the numbers indicate timing, whereas the explanation in the text relates to duration. Please justify.

Discussion:

1. It is better to interpret the discussion in one context.

2. In the participant characteristics, why is the DSRS-c lower than in the previous studies? Given the difference in time and era, you need to justify the possible causes.

3. It is helpful to include studies that examine the depressive symptoms from childhood to early adulthood to exclude confounding factors

Conclusion:

The conclusion is comprehensive. In line 455, further studies are needed before concluding that poor social interactions cause depressive symptoms and decreased physical activity time.

It is favourable to add a recommendation in this regard.

6. PLOS authors have the option to publish the peer review history of their article (what does this mean?). If published, this will include your full peer review and any attached files.

Reviewer #1: No

Reviewer #2: No

Reviewer #3: No

---

## [Author Response · Author response to Decision Letter 0]

15 Apr 2024

We sincerely thank you and the reviewers for their careful assessment of our manuscript and for their constructive and suggestive comments, which have helped improve the overall quality of our manuscript. Point-by-point responses to all comments by the academic editor and reviewer's have been prepared and provided below. 

We hope that these revisions are sufficient to make our manuscript suitable for publication in PLOS ONE and look forward to hearing from you at your earliest convenience.

----------

Response to Academic Editor

Academic Editor (1): Ensure that your manuscript meets PLOS ONE's style requirements. 

Response (1): We have carefully checked PLOS ONE's style requirements and modified the typeface of Heading 3 to match the template. We checked the file name carefully prior to uploading the file.

Academic Editor (2): We notice that your supplementary figures are uploaded with the file type 'Figure'. Please amend the file type to 'Supporting Information'.

Response (2): In accordance with the journal requirements, the file name of the supplementary figure was changed to 'supporting information' and uploaded.

----------

Response to Reviewer #1

Thank you for taking the time to review this manuscript. The reviewer's comment was, "Overall, the manuscript appears to be well-prepared and meets the standards for publication." Based on this comment, no particular points needed to be addressed.

The evaluation of Data Availability was "No", but we understand that this is due to the fact that "No-some restrictions will apply" was stated in our Data Availability section. The data for the current study is not publicly available, but can be provided upon reasonable request and with the approval of the relevant department in Koshu City.

----------

Response to Reviewer #2

Thank you for evaluating each component of this manuscript. Overall, the comments were positive, and no particular points were needed to be addressed.

----------

Response to Reviewer #3

We appreciate your detailed comments and advice to improve the overall quality of this manuscript. We believe that the unclear descriptions are now improved. We have provided relevant explanations for all incorporated revisions.

Introduction (1): The introduction provides a comprehensive background on depression in adolescents. In line 37, the included study with 3331 children was conducted in 2004, and the other research (line 41) was conducted in 2016. It would be better to connect this information as a trend. This will enable the reader to comprehend the trend of depression over the years.

Response - Intro. (1): We agree that the different time points at which the two articles were reported are crucial information. In line 38, we stated that the study in [2] was conducted in 2004 and that the DSRS-C scores of Japanese children are consistent in lines 42-43.

Introduction (2): The percentage of depression in the DSR-C studies was mentioned. In line 43, the percentages of depression were more significant in Europe and the USA. However, providing more specific information by including figures would be helpful. This would contextualise the prevalence of depression locally and worldwide.

Response - Intro (2): We agree that stating the relevant values will enhance the reader's understanding. Accordingly, we have stated the report of Ivarsson et al. (1994) in lines 44-46 as an example in the revised version of the manuscript.

Introduction (3): The symptoms of depression were mentioned in the second paragraph - however, the sentence in line 50 needs paraphrasing to deliver the intended point.

Response - Intro (3): We agree that the clarity of the original statement was poor and warranted clarification. Accordingly, this statement was rephrased to improve the clarity and readability (lines 51-54). Thank you for highlighting this issue.

Introduction (4): In this third paragraph, the recommended duration of physical activity is documented. However, it would also be better to mention the recommended exercise type. (frequency and intensity). This will provide a whole picture of the physical activity recommendation.

Response - Intro (4): The wording to indicate the recommended types and frequency of exercise was hard to understand and has been corrected in lines 62-70. Note that the Japanese guideline introduces a diverse range of physical activity types and, therefore, does not provide recommended conditions for specific exercise intensity. This point was reinforced by the WHO recommendation.

Introduction (5): In the last paragraph, bukatsu sport was mentioned. It would be helpful to briefly describe this sport and say if it is optional or mandatory for the students. This will keep the reader on track.

Response - Intro (5): Thank you for highlighting this issue. We agree that for those who do not live in Japan, imagining the specifics of bukatsu could be difficult, warranting the addition of further information. In lines 87-88, we emphasize that although bukatsu is not mandatory, many students participate in it. Additionally, we have listed the names of specific sports in bukatsu in line 91.

Methods (1): This study includes massive and structured data, making the results more reliable. However, it was initially challenging to comprehend whether the study was retrospective or both retrospective and prospective. For example, In the 3rd paragraph, the author explains that a self-administered questionnaire was given to the participants. Was this done by the Koshu project or from your side by interviewing the subjects? Please clarify these points to be easier for the reader to follow.

Response - Methods (1): This study was a prospective assessment, obtaining and analyzing cohort data conducted by the Koshu Project. The wording in the third paragraph was unclear; hence, the content was revised in lines 133-137.

Methods (2): In the participants and procedure, the data were collected from the Koshu project. Out of these, 1890 were selected. Does this represent all the available data, or was a sample size calculated? Please specify this information to comprehend the methodology.

Response - Methods (2): The 1,890 represents the number of all available data. We have modified the relevant statement (line 124) to indicate that this value indicates the overall data.

Methods (3): Under the section depressive symptoms, some of the studies used 15 as a cut-off point. Please provide a more substantial justification if possible. This might underestimate the depressive symptoms, and it would be difficult to compare it with other international studies.

Response - Methods (3): Even if the same scale is employed, different evaluation criteria will warrant careful discussion of the results; however, our manuscript did not consider this point. The rationale for setting the cutoff point at 16 is clarified in lines 148-153.

Methods (4): Does the physical activity time mean the duration or the timing? In the methods, the question refers to the duration; however, the table refers to the timing, as there is another question for the activity based on the duration spent in the physical activity. The world time is, therefore, overlapping between the timing and the duration.

Response - Methods (4): Physical activity time represents the duration for which physical activity was performed during the week. In Table 1, [hours:minutes] was used to represent the duration of physical activity time, but since the question asked " How many hours a week do you usually spend on physical activities... " and the decimal point was used in the text as well, the minutes were unified to be represented by a decimal point.

Methods (5): Figure 1 shows the recruitment done in this study, which is a great visual explanation. However, it must be self-explanatory to make it easier for the reader to follow.

Response - Methods (4): The wording in the top box of Fig 1 was modified to clearly indicate that the data represent ALL available participants.

Methods (6): Table 1 divides body image into wanting to be fat, wanting to stay the same and wanting to be thin. However, the text line 172 mentioned that it would be classified into two categories: want to be thin and don't want to be thin. Please change the results in the table to be consistent.

Response - Methods (6): As pointed out, "want to be fat" and "want to stay the same" in Table 1 were merged and classified into two categories: "want to be thin" and "do not want to be thin."

Methods (7): Table 1, including physical activity time. Please specify if it is referred to as timing or duration, as the numbers indicate timing, whereas the explanation in the text relates to duration. Please justify.

Response - Methods (7): Point 4 above clarifies the implication of physical activity time. The wording in Table 1 has been modified from [hours:minutes] to a notation of time, including a decimal point.

Discussion (1): It is better to interpret the discussion in one context.

Response - Discussion (1): To comprehensively understand the results of this study, we thought it would be appropriate to discuss the results of the sub-analysis before discussing the main analysis results; hence, we structured the argument in this manner. A relevant explanation has been provided in lines 336-338.

Discussion (2): In the participant characteristics, why is the DSRS-c lower than in the previous studies? Given the difference in time and era, you need to justify the possible causes.

Response – Discussion (2): It is difficult to explain structurally why the rate of depressive tendencies among the participants in this study is lower than in other previous studies in Japan based on the data of this study alone. Nevertheless, in lines 308-313, we discuss the potential of the social environment in Koshu City and the duration of physical activity, which is discussed later. Regarding the difference in time periods noted by reviewer 3, we explained in the Introduction (lines 42-43) that this is consistent in Japan.

Discussion (3): It is helpful to include studies that examine the depressive symptoms from childhood to early adulthood to exclude confounding factors

Response - Discussion (3): While some studies have found that age factors are not associated with depressive symptoms in adolescents [38], a 2018 Japanese study [32] identified significant differences by age, which may be a confounding factor. The current study did not assess this effect itself because it excluded data from children with depressive tendencies at age 11. This has been explained in the Limitations of the Study section (lines 425-430). Relevant references were added to the list of references [38].

Conclusion: The conclusion is comprehensive. In line 455, further studies are needed before concluding that poor social interactions cause depressive symptoms and decreased physical activity time. It is favourable to add a recommendation in this regard.

Response - Conclusion: As pointed out, further research is needed to comprehensively elucidate these relationships, and we have stated the need for such research in lines 453-456.

----------

Some additional modification: Some spelling errors and inappropriate wording were found and corrected.

---

## [Decision Letter · Decision Letter 1]

13 May 2024

PONE-D-23-34588R1Association between physical activity time and depressive symptoms in adolescents: A longitudinal study in a rural city in JapanPLOS ONE

Dear Dr. Kawai,

Thank you for submitting your manuscript to PLOS ONE. After careful consideration, we feel that it has merit but does not fully meet PLOS ONE’s publication criteria as it currently stands. Therefore, we invite you to submit a revised version of the manuscript that addresses the points raised during the review process.

We look forward to receiving your revised manuscript.

Kind regards,

Mukhtiar Baig, Ph.D.

Academic Editor

PLOS ONE

Journal Requirements:

Reviewers' comments:

Reviewer's Responses to Questions

**Comments to the Author**

1. If the authors have adequately addressed your comments raised in a previous round of review and you feel that this manuscript is now acceptable for publication, you may indicate that here to bypass the “Comments to the Author” section, enter your conflict of interest statement in the “Confidential to Editor” section, and submit your "Accept" recommendation.

Reviewer #3: (No Response)

2. Is the manuscript technically sound, and do the data support the conclusions?

Reviewer #3: Yes

3. Has the statistical analysis been performed appropriately and rigorously? 

Reviewer #3: Yes

4. Have the authors made all data underlying the findings in their manuscript fully available?

Reviewer #3: Yes

5. Is the manuscript presented in an intelligible fashion and written in standard English?

Reviewer #3: Yes

6. Review Comments to the Author

Reviewer #3: I wanted to extend my gratitude for the opportunity to review your paper. It was a privilege to engage with your work. I'm pleased to inform you that I have completed my review and have provided some suggestions and corrections that I believe will further strengthen your paper.

Abstract:

In line 22, physical activity was abbreviated as PA; however, in line 24, the full sentence was used. Please choose whether to consistently use the abbreviation or the full term. This will ensure writing consistency and provide clarity of purpose for the abbreviation.

Introduction:

• The term "physical activity" is written in full in line 57, 75, 76, and elsewhere. Please use the search function to replace the full term with "PA" if you decide to keep the abbreviation.

• In line 70, please leave a space between "5-17" and "years old."

• It is recommended to define "Bukatsu" immediately after its introduction in line 68 rather than later in line 72 to maintain reader comprehension.

Methods:

• Please mention ethical considerations either at the beginning or end of the methods section.

• Clarify whether both retrospective (lines 103-139) and prospective approaches (lines 143-150) were used.

• Also, specify the number of schools visited and the total sample size from these schools. If the sample size is included from the1890, please indicate this.

Physical Duration:

• In some contexts, "time" and "duration" can be used interchangeably, but they don't always mean the same thing and might cause confusion. "Time" generally refers to a specific point, while "duration" specifically refers to the length of time that something lasts. Therefore, the word "duration" would be more suitable in the context of your research.

Results:

• Please specify physical activity duration in the table as hours and minutes.

• Consider changing the word "time" to "duration" to eliminate confusion for the reader.

7. PLOS authors have the option to publish the peer review history of their article (what does this mean?). If published, this will include your full peer review and any attached files.

Reviewer #3: No

---

## [Author Response · Author response to Decision Letter 1]

13 May 2024

Response to Reviewer #3

We appreciate your detailed comments and advice. We believe that your suggestion improves the overall quality of this manuscript.

Abstract: In line 22, physical activity was abbreviated as PA; however, in line 24, the full sentence was used. Please choose whether to consistently use the abbreviation or the full term. This will ensure writing consistency and provide clarity of purpose for the abbreviation.

Response - Abstract: Unified to full sentence (physical activity).

Introduction (1): The term "physical activity" is written in full in line 57, 75, 76, and elsewhere. Please use the search function to replace the full term with "PA" if you decide to keep the abbreviation.

Response - Introduction (1): In the whole manuscript, "physical activity" was used as a complete term. In the figures and tables, the full term was also used in basic. Abbreviations (PA) were used due to space limitations; explanations were provided outside the tables and in the captions.

Introduction (2): In line 70, please leave a space between "5-17" and "years old." 

Response - Introduction (2): Space was inserted in line 68.

Introduction (3): It is recommended to define "Bukatsu" immediately after its introduction in line 68 rather than later in line 72 to maintain reader comprehension. 

Response - Introduction (3): Moved the description part of the bukatsu to one-sentence before (lines 88-90).

Methods (1): Please mention ethical considerations either at the beginning or end of the methods section. 

Response - Methods (1): Ethical considerations were described in the Participants and Procedure section, including the approval number of the ethical review (lines 144-146).

Methods (2): Clarify whether both retrospective (lines 103-139) and prospective approaches (lines 143-150) were used.

Response - Methods (2): Described a retrospective analysis to examine physical activity patterns and depressive symptoms from 11 y to 14 y, and a prospective analysis to observe progression of depressive symptoms and physical activity patterns in lines 131-135.

Methods (3): Specify the number of schools visited and the total sample size from these schools. If the sample size is included from the1890, please indicate this.

Response - Methods (3): All public elementary and junior high schools (13 elementary schools and 5 junior high schools) in Koshu City are participating in the project. The number of each school is included in line 130. The project was conducted by the Koshu City Board of Education, and the researchers themselves did not visit the schools during the survey. The flow diagram of the sample used for the analysis from the 1890 responses is shown in Fig. 1.

Methods (4): Physical Duration - In some contexts, "time" and "duration" can be used interchangeably, but they don't always mean the same thing and might cause confusion. "Time" generally refers to a specific point, while "duration" specifically refers to the length of time that something lasts. Therefore, the word "duration" would be more suitable in the context of your research. 

Response - Methods (4): Unified to Physical activity "duration" instead of physical activity "time." The title of this manuscript was also changed from "time" to "duration."

Results (1): Please specify physical activity duration in the table as hours and minutes.

Response - Results (1): The notation was reverted to hrs:min and the units are given in the table.

Results (2): Consider changing the word "time" to "duration" to eliminate confusion for the reader. 

Response - Results (2): The terminology was changed from "time" to "duration" in all parts of this manuscript.

---

## [Editor Report · Decision Letter 2]

20 May 2024

Association between physical activity duration and depressive symptoms in adolescents: A longitudinal study in a rural city in Japan

PONE-D-23-34588R2

Dear Dr. Kawai,

We’re pleased to inform you that your manuscript has been judged scientifically suitable for publication and will be formally accepted for publication once it meets all outstanding technical requirements.

Kind regards,

Mukhtiar Baig, Ph.D.

Academic Editor

PLOS ONE

---

## [Editor Report · Acceptance letter]

22 May 2024

PONE-D-23-34588R2 

PLOS ONE

Dear Dr. Kawai, 

I'm pleased to inform you that your manuscript has been deemed suitable for publication in PLOS ONE. Congratulations! Your manuscript is now being handed over to our production team.

Kind regards, 

on behalf of

Professor Mukhtiar Baig 

Academic Editor

PLOS ONE